Exceptionally preserved insect fossils in the Late Jurassic lagoon of Orbagnoux (Rhône Valley, France)

Nel André 1 anel@mnhn.fr pnel@mnhn.fr
Nel Patricia 1
Krieg-Jacquier Régis 2
Pouillon Jean-Marc 3
Garrouste Romain 1 garroust@mnhn.fr
1 Muséum National d’Histoire Naturelle, Institut de Systématique, Evolution, Biodiversité, ISYEB , CNRS UMR 7205, UPMC EPHE, Paris , France
2 Barberaz , France
3 Nivolas Vermelle , France
De Baets Kenneth
Electronic publication date: 2014 Sep 2
Publication date: 2014
Volume: 2
Electronic Location ID: e510
Received 2014 Jun 15; Accepted 2014 Jul 21
Copyright: © 2014 Nel et al.
Copyright year: 2014
Copyright holder: Nel et al.
License: This is an open access article distributed under the terms of the Creative Commons Attribution License, which permits unrestricted use, distribution, reproduction and adaptation in any medium and for any purpose provided that it is properly attributed. For attribution, the original author(s), title, publication source (PeerJ) and either DOI or URL of the article must be cited.
License URL: https://creativecommons.org/licenses/by/4.0/

Keywords: Insecta, Heteroptera, gen. et sp. nov., Attacks on Zamites leaves, Mesoveliidae, Trails in sediment

Funding: CNRS UMR 7205 This paper was funded by the CNRS team UMR 7205. The funders had no role in study design, data collection and analysis, decision to publish, or preparation of the manuscript.

==============================
The Late Kimmeridgian marine limestones of the area around Orbagnoux (Rhône, France) are well known for their fish fauna and terrestrial flora. Here we record the first insects and their activities (mines on leaves and trails in sediments) from these layers, including the oldest record of the gerromorphan bugs, as a new genus and species Gallomesovelia grioti, attributed to the most basal family Mesoveliidae and subfamily Madeoveliinae. These new fossils suggest the presence of a complex terrestrial palaeoecosystem on emerged lands near the lagoon where the limestones were deposited. The exquisite state of preservation of these fossils also suggests that these outcrops can potentially become an important Konservat-Lagerstätte for the Late Jurassic of Western Europe.

Introduction

Tithonian Konservat-Lagerstätte of lithographic limestone of Bavaria is well known with the numerous discoveries of emblematic fossils (Archaeopteryx, etc.) (Schweigert, 2007), clearly fewer fossils were obtained from the French Late Kimmeridgian equivalents of these rocks in the departments of Ain and Rhône. Only the lithographic limestone formation of Cerin has been the subject of significant scientific studies on taphonomy, flora, and ichnofossils (Bernier et al., 1991; Bernier et al., 2014). The fauna is essentially of marine origin (Wenz et al., 1994; Gaillard et al., 2006) if the presence of vertebrate trackways demonstrates that some terrestrial environments were very close to this place. Neither insect nor trace was discovered at Cerin, unlike the rich palaeoentomofauna of the lithographic limestone of Bavaria (Carpenter, 1992). The sites around Orbagnoux were better candidates for the discovery of terrestrial arthropods because fossil plants are more frequent and well preserved than in other places like Cerin (Barale, 1981). However, early exploration of this area during the 19th and 20th century contributed to our knowledge on Jurassic fossils but no new studies have been carried out in more recent times.

During two field works during 2012 and 2013, with the help of the Société des Naturalistes et Archéologues de l’Ain and the Groupe ‘Sympetrum’ (Recherche et Protection des Libellules), we could discover in the outcrops around Orbagnoux a small bug that represents the first Late Jurassic insect of France, plus insect-mediated plant damages, and probably track ways of aquatic fly larvae. These first discoveries suggest that more terrestrial arthropods might be present in these layers, although these fossils seem to have been washed in the lagoonal environment. Also the quality of preservation of these fossils is better than those of the fossil insects from the Bavarian lithographic limestone, suggesting that Orbagnoux can become one of the major insect outcrops for the Late Jurassic.

Material and Methods

The fossils were collected in the small valley of the river Dorches (a river Rhône tributary), situated on the territory of the Corbonod village, North West of Orbagnoux mine (Long. 5°46′ 32.2″E, Lat. 45°59′ 26.3″N, alt. 597 m), and in the newly recorded outcrop of the “Croix de Famban” (situated along the small road D123, alt. 1310–1313 m, Long. 5°45′ 58.5″E, Lat. 45°56′ 58.9″N, and along the eastern margin of the anticlinal surface, 200 m south of the road), situated south-west of Orbagnoux, in lithographic and bituminous limestones of the same horizon (see Fig. 1). These levels are raised and follow the anticlinal of le Grand Colombier situated west parallel to the Rhône valley between Corbonod and Songieu (Gudefin, 1968). Tribovillard et al. (1999) considered that this formation is very poor in macroorganisms (“quasi-abiotic platy limestone accumulation”). The highly bituminous black shales inside of the Orbagnoux mine (Chateauneuf et al., 1982) contain few macrofossils (A Nel, pers. obs., 2012): mainly rather poorly preserved ammonites and bivalves, foraminifers and ostracods. On the contrary, fossil plants, fishes and crustaceans are known in other layers outside of the mine (Barale, 1981; Tejo Yuwono, 1987) (see Fig. 2). More precisely, plants remains (leaves, fruit cones, seeds, branches, pollen) are very frequent in the less bituminous dark brown levels outside of the mine in the Dorches valley. The flora yielded almost 34 species which belong to plant groups quite well known from the Jurassic: Pterydophytes, Pteridospermales, Cycadales, Ginkgoales, Bennetitales, and Coniferales (Barale, 1981). The most frequent leaves correspond to bennetitalian foliage type Zamites sp. Fossil fishes (Sauvage, 1893; Wenz et al., 1994), vertebrate coprolites, crustaceans, oysters, small ammonite shells and aptychi (Enay et al., 1994) can be found in yellow lithographic layers situated three meters above the plant layers (Barale, Philippe & Thevenard, 1992). An insect has been found in the same yellow lithographic layers; it is described below. On the contrary the animals seem to be quite rare in the plant layers. A second small outcrop is situated 30 m (alt. 657 m, along a small forest road) south of the main one. It is very rich in plant remains, some with insect attacks, described below. The new outcrop of the “Croix de Famban” (unrecorded in Barale, 1981) has given also thick layers rich of large leaves of Zamites and delicate laminites with small fishes, crustaceans, and traces of animal activities in the sediment (some being herein attributed to fly larvae).

Figure 1 Geographic map with locations of the three concerned outcrops.

Figure 2 Fauna.

Selected fossil organisms from Orbagnoux outcrop. (A) Brachyphyllum elegans, specimen JMP 455 (coll. J-M Pouillon); (B) Pachypterix sp., specimen JMP 391 (coll. J-M Pouillon); (C) Leptolepis cf. voithi (coll J-C Demaury); (D) Crustacea Penaeidae (coll J-C Demaury) (scale bars 10 mm).

These lithographic and bituminous limestones (Late Kimmeridgian “calcaire en plaquettes” Formation) correspond to a marine shallow lagoonal environment. The formation of these carbonate deposits are due to the development of microbial mats (“kopara” Tribovillard et al., 1999; Tribovillard et al., 2000). These authors indicated that the palaeoenvironment corresponds to a lagoon delimited by a reef barrier and to islands occupied by terrestrial organisms (plants and animals) coming into the lagoon. Some insects (fly larvae) could apparently live in the shallow brackish sediments (see below).

Specimen examinations have been made with an Olympus microscope SZX9 and drawings made with a drawing tube. Environmental SEM images have been made with Tescan Vega II LSU in variable pressure mode (SE, secondary electron).

The electronic version of this article in Portable Document Format (PDF) will represent a published work according to the International Commission on Zoological Nomenclature (ICZN), and hence the new names contained in the electronic version are effectively published under that Code from the electronic edition alone. This published work and the nomenclatural acts it contains have been registered in ZooBank, the online registration system for the ICZN. The ZooBank LSIDs (Life Science Identifiers) can be resolved and the associated information viewed through any standard web browser by appending the LSID to the prefix “http://zoobank.org/”. The LSID for this publication is: urn:lsid:zoobank.org:act:pub:8154DAD2-355D-4591-A4FC-7E6EFC95B368. The online version of this work is archived and available from the following digital repositories: PeerJ, PubMed Central and CLOCKSS.

Systematic Paleontology

Adult insect from Orbagnoux

Infraorder Gerromorpha Popov, 1971	
Family Mesoveliidae Douglas and Scott, 1867	
Subfamily Madeoveliinae Poisson, 1959	
Included genera. Modern Madeovelia Poisson, 1959 and Mesoveloidea Hungerford, 1929, Gallomesovelia gen. nov.	

Genus Gallomesovelia gen. nov.	
Type species. Gallomesovelia grioti sp. nov.	
Diagnosis. Adult macropterous characters. Large size (body 6.0 mm long);
large rounded eyes; tegmen covered with setae; abdominal sternite 8 similar to sternite 7.	
Etymology. Named after Gallia, ancient Latin name for France,
and Mesovelia. Gender feminine.	

Gallomesovelia grioti sp. nov.	
(Fig. 3)	
Material. Holotype specimen MNHN.F.A51098
(print and counterprint of a nearly complete bug, fossilised in lateral view,
legs and antennae not preserved).	
Diagnosis. As for the genus	
Etymology. Named after our friend and colleague Claire Griot,
who rendered possible the field researches around Orbagnoux outcrop.	

Remark. A rather large (ca. 6 cm long) shrimp is fossilised in the piece of rock 0.5 mm (three laminae) below the print of this bug. It was first detected by the presence of a depression in the sediment and is only evident on the x-radiograph (see Fig. 4).

Figure 3 Mesoveliidae.

Aquatic bug from Orbagnoux outcrop: Gallomesovelia grioti sp. nov. (Holotype MNHN.F.A51098). (A) reconstruction; H, head; E, Eye; M, mesonotum; Pr, pronotum; As, abdominal segments; Cx, coxae; Me, metanotum; Mee, metanotum elevation; Wv, wing veins; Gs, genital segment; Pt, pterostigma; (B) photograph without alcohol; (C) photograph under alcohol (scale bars 1 mm for A, B and C).

Figure 4 Crustacea.

Crustacea indet., Malacostraca? X-ray photograph of a specimen fossilised below holotype of Gallomesovelia grioti (scale bar 10 mm).

Description. Body 6.0 mm long; with wings and thorax covered with a distinct pile of microsetae; head 0.75 mm long, 0.87 mm high, deflected in front of eyes but without transverse constriction; compound eye rather large, rounded, 0.35 mm diameter; no ocelli; cephalic trichobothria not preserved; insertion of antenna visible below eye; labium elongate, reaching level of mid coxae, four-segmented, with segments 1 and 2 very short, segment 3 very long, 1.32 mm long, much longer than segment 4, 0.37 mm long; no pair of bucculae covering base of labium; thorax 2.2 mm long; prothorax 0.62 mm long 3.8 mm high, with a suture separating tergum and sternopleuron; pronotum short; mesoscutellum triangular exposed, 0.5 mm long; a small metanotal elevation; forewing 3.6 mm long, covered with microsetae, lacking claval commissure but with two basal cells and one apical cell and a “pterostigma”-like broadening of vein Sc + R (subcosta + radius); coxae inserted close to ventral midline of thorax, all oriented obliquely backward (especially for the posterior ones), distance between fore and mid coxae 0.54 mm, between mid and hind coxae 1.0 mm; abdomen 3.1 mm long, ca. 1.7 mm high, with seven sternites visible laterally, i.e., sternites 2 to 7 plus sternite 8 narrower than other sternites, and not divided in two parts (male).

Discussion. This fossil bug falls in the Gerromorpha because of the following characters, after Schuh & Slater (1995): compound eyes large, rounded; head not constricted transversely; forewing lacking claval commissure, not divided into a corium-clavus-membrane; wings and part of body covered with a distinct pile of microsetae. Unfortunately the cephalic trichobothria, characteristic of the Gerromorpha, are not visible above the eye (Andersen, 1982). An attribution to the subfamily Madeoveliinae would be supported by the following characters: macropterous; mesoscutellum triangular exposed; coxae inserted close to the ventral midline of thorax (plesiomorphy); no pair of bucculae covering base of labium; ocelli lacking; forewing with two basal cells and one apical cell (Andersen, 1982; Andersen, 1999; Schuh & Slater, 1995).

Male genitalia and egg characters, important in the family diagnosis are unfortunately not detected in the type specimen of Gallomesovelia gen. nov. Nevertheless, the mesoscutellum present but relatively reduced and the metanotal elevation, are both apomorphic characters present in Gallomesovelia, the macropterous Mesoveliidae but also in the Hebridae (Damgaard, 2008b: 454).

The attribution of Gallomesovelia to the Madeoveliinae is supported by the following apomorphies (Andersen & Polhemus, 1980; Andersen, 1999; Damgaard et al., 2012: 193): head deflected in front of eyes; adults macropterous and without ocelli. More precisely, the absence of ocelli supports an attribution to the Madeoveliinae, as the reduction of ocelli in the macropterous morphs is convergent in the Mesoveliidae: Madeoveliinae, the Gerridae, plus some Veliidae and Hydrometridae (Damgaard, 2008b; Damgaard et al., 2012). The two modern madeoveliine genera Madeovelia and Mesoveloidea have the plesiomorphic presence of prothoracic suture separating tergum and sternopleuron (Andersen, 1999: 13; Damgaard et al., 2012: 193). Gallomesovelia has this character too. The shape of the pronotal lobe cannot be determined because of the animal is visible in lateral view; nevertheless, it seems that there is no anterior constriction as in winged Mesoveliinae (Andersen & Polhemus, 1980). The general shape of body and forewing venation of Gallomesovelia are similar to those of Madeovelia and Mesoveloidea in the large rounded eyes, shape of labium, tegmen covered with setae (Poisson, 1959; Hungerford, 1929; Jaczewski, 1931; Moreira, Ribeiro & Nessimian, 2006). The main difference of Gallomesovelia with both the male and the female of Mesoveloidea is the abdominal sternite 8 as large and long as sternite 7. A further important difference is the greater size (body 6.0 mm long in Gallomesovelia instead of 2.9 mm in Mesoveloidea peruviana Drake, 1949, 2.9–3.8 mm in Mesoveloidea williamsi Hungerford, 1929, and 2.6 mm in Madeovelia guineensis Poisson, 1959) (Hungerford, 1929; Poisson, 1959; Schuh & Slater, 1995). More generally Gallomesovelia is rather large compared to the modern and fossil Mesoveliidae.

The Mesoveliidae are considered as the sister group to all other Gerromorpha (Damgaard, 2008a; Damgaard, 2008b; Weirauch & Schuh, 2011; Damgaard et al., 2012), but not the oldest Gerromorpha + Panheteropera known (Panheteroptera = Nepomorpha + Leptodomorpha + Cimicomorpha + Pentatomorpha). The oldest known Heteroptera are undescribed Nepomorpha from the Triassic (Late Carnian) of Virginia (Fraser & Grimaldi, 1999).

Gallomesovelia is the oldest known representative of the superfamily Gerromorpha. Li et al. (2012) placed the gerromorphan radiation during the Jurassic using molecular phylogeny and fossil calibration. Our fossil is congruent with all the current hypotheses.

The Gerridae, Hydrometridae, and Veliidae are known in the Early Cretaceous (Solòrzano-Kraemer et al., 2014), while Hebridae are only recorded from the Cenozoic (Damgaard, 2008a). Thus the absence of the Mesoveliidae in the Mesozoic remained surprising. The present discovery confirms the recent attributions of some Cretaceous fossils to this family: the immature gerromorphan nymph from the Mid Cretaceous French amber described by Perrichot, Nel & Néraudeau (2005) is currently attributed to the Mesoveliidae as one of us (AN) suggested in the first version of the paper of 2005, while two other Mesoveliinae are also newly recorded from the same amber (Solòrzano-Kraemer et al., 2014). Nevertheless all the other Mesozoic taxa previously attributed to the Mesoveliidae are currently considered as Heteroptera incertae sedis (Damgaard et al., 2012). The Miocene Mesovelia dominicana Garrouste & Nel, 2010 remains the unique described Cenozoic Mesoveliidae.

Insect herbivory on Zamites leaves

Although traces of insect herbivory are rather frequent and diverse in the fossil record (Labandeira & Currano, 2013), the Late Jurassic is a period with a relatively poor record of such traces, compared to the Triassic or to the end of the Lower Cretaceous (Labandeira, 2006). Therefore any new fossil, like the present discoveries, is welcome. Jud, Rothwell & Stockey (2010) listed evidences of ‘cone boring, anthophily (pollination), pollenivory, wood borings, cortex borings, leaf oviposition, leaf grazing, and leaf galling’ but no insect mines on the Mesozoic Bennettitales. Pott et al. (2008) described oviposition damages on foliage from the Upper Triassic of Austria. Pott et al. (2012) and Edirisooriya & Dharmagunawardhane (2013) described leaf-margin feedings on Middle Jurassic Anomozamites villosus and Otozamites beanie respectively. Schweigert & Dietl (2010) described rounded structures on Zamites feneonis as “probable prints of former galls” from the Late Kimmeridgian of the Nusplinger lithographic limestone in Germany. Harris (1942) also described galls on the bennettitalean Anomozamites.

The first found fossil (specimen MNHN.F.A51100, A. Nel leg.) is the print and counterprint of a leaf of a Zamites species, 70.0 mm long, 75.0 mm wide, with 13 leaflets preserved, with several surface feeding traces. These structures are longitudinally oriented, generally situated along the margins of the leaflets, elongate, 5.0–16.0 mm long, 2.0–2.5 mm wide. The sides of the slots are straight and the ends are rounded. Their central parts are devoid of organic matter in the print but the surface of the leaflet cuticle is still present in the counterprint. They are surrounded by a darker zone that corresponds to a rim of reaction tissue 0.5–1.0 mm wide, darker than the organic matter of the main part of the leaflet (see Fig. 5). Except in one case, there is only one attack per leaflet and only seven of the 13 leaflets are attacked.

Figure 5 Zamite attack 1.

Insect traces on a Zamites leave (MNHN.F.A51100). (A) print, Orbagnoux outcrop; (B) counterprint (scale bars 10 mm).

Two other Zamites leaves with traces of insect activities were found in the collection François Escuillié (Gannat, France), coming from the same outcrop and level (Fig. 6). The second specimen is a large leaf with all leaflets attacked by insects, some of them covering nearly all the surface of the leaflet. Nevertheless they are identical to the traces on the first leaf (see Fig. 5). The third specimen is a unique small hyaline oval trace filled with carbon and looks slightly different from the traces on the first and second specimens.

Figure 6 Zamites attack 2.

Insect traces on Zamites leaves, specimens from Escuillié collection. (A) specimen with nearly all leaflets attacked (scale bar 2 cm), (B) second specimen with only one attack on a leaflet (scale bar 1 cm).

Although the leaves of Zamites are very frequent in the outcrop of Orbagnoux, no other trace of insect activity was detected among ca. 100 leaves of Zamites we collected there. They can be considered rare. No other plant was found attacked.

The evidence from insect interactions with bennettitalian plants from Orbagnoux is completely different from the structures already described on other Zamites leaves, even to those described on some Upper Triassic Zamites from the United States (Ash, 1996: 242, figures 3–4, Ash, 2005).

The structures from Orbagnoux could correspond to a surface feeding DT103 “elongate window feeding (sub)parallel to major venation” type (sensu Labandeira et al., 2007), rather than to mining activities as the surface of the leaflet cuticle is preserved at least on one side and no remains of coprolite were found.

The main differences with the damages on Triassic Zamites described by Ash (1996) are as follows: those from Orbagnoux are broader, covering more than two veins of the leaflet; they are situated along the leaflet margin instead of being 1–3 mm from the margins. Thus they were probably caused by different insects. Ash (1996) supposed that the Triassic attacks could have been caused by “grazing insects”. It is quite delicate to determine which type of insect could have caused the mines from Orbagnoux. Nevertheless, a holometabolous larva could be responsible of such attacks. Similar attacks are frequent on modern angiosperm leaves, caused by larvae of Diptera, Lepidoptera, or Coleoptera (A Nel, pers. obs., 2014). Because of the hardness of the Zamites leaflets, it is more likely a beetle larva that could have caused these attacks. These structures are similar to traces produced by modern Chrysomelidae, but the presence of Chrysomelidae during the Jurassic is still debated (compare Gómez-Zurita et al., 2007; Wang et al., 2014).

Sinusoidal trails similar to trace fossil Cochlichnus Hirchock, 1858

These sinusoidal traces of an animal activity were found in thin yellow lithographic laminites in the outcrop of the “Croix de Famban” (specimen MNHN.F.A51099, Fig. 7). Small fishes, vertebrate coprolites, worm trails, and crustaceans (isopods and shrimps) are also present in the same layers. Plants are frequent in thicker layers just above these laminites but absent in the laminites. These traces are similar to the modern grazing trails formed by stratiomyid or ceratopogonid fly larvae in shallow freshwater or brackish sediments (Metz, 1987; Mángano, Buatois & Claps, 1996). The oldest known Stratiomyidae and Ceratopogonidae are Lower Cretaceous (Huang & Lin, 2007; Borkent, Coram & Jarzembowski, 2013). Similar traces in marine environments have been interpreted of worm- or gastropod-origin (Hakes, 1976; Dam, 1990). Together with the other organisms found in this outcrop, they demonstrate that the corresponding palaeoenvironment was clearly not abiotic, as postulated by Tribovillard et al. (1999).

Figure 7 Trackways.

Sinusoidal trails cf. Cochlichnus (Croix de Famban outcrop, MNHN.F.A51099), (A) print, (B) counterprint (scale bars 10 mm).

Palaeoenvironmental implications

The main part of the fauna found in the Late Jurassic layers around Orbagnoux clearly correspond to marine environments (ammonites, bivalvia, shrimps, isopods, fishes, etc.), the present discoveries confirm the presence of a very close terrestrial biota, which was supposed after the presence of a rather diverse flora represented by leaves and reproductive organs (cones). The rarity of insects together with the partly decayed state of the type specimen of Gallomesovelia grioti (legs and antennae missing), suggest a transport of these terrestrial organisms into the marine environment. Modern mesoveliids (water treaders) are predators, and strictly freshwater or terrestrial insects, living in humid places, but some are associated with marine habitats (intertidal). Gallomesovelia grioti could have lived very close to the sea margin or even in the brackish part of the lagoon itself, as some modern Mesoveliidae (Andersen, 1982: 338: fig. 597): it was maybe a surface skater because of the particular orientations of the coxae, directed backwards (Andersen, 1982: 288–289). Due to the preservation of the terrestrial insect bodies that is generally rather poor in the Bavarian Tithonian (Tischlinger, 2001), it would be nearly impossible to expect to find there such small and delicate insects similar to Gallomesovelia grioti. This situation renders very precious any new discovery of terrestrial arthropods in Orbagnoux outcrops. The discovery of the traces of insect activities on a Zamites leaf shows that the diversity of the terrestrial arthropods was probably significant, the insect fauna not being reduced to few aquatic or subaquatic bugs, like in a simple atoll lagoon beach or beach rock. Nevertheless future investigations shall be necessary to discover the palaeodiversity of this new insect world.

We thank Vladimir Makarkin and two anonymous referees for their valuable comments on the first version of this paper. We sincerely thank the Société des Mines d’Orbagnoux for the kind authorization to collect fossils in their property. Special thanks to Torsten Wappler for information and references on insect herbivory on Zamites. We also thank the following colleagues for their kind help during the field campaigns in 2012 and 2013: Bernard Béreyziat, Michel Béreyziat, Guy Robert, Pierre Roncin, Jean-Marie Demaury, Thierry Virton, Bernard Janvier, and Daniel Grand. A special thanks to Claire Griot because she rendered possible these investigations, inviting us into her wonderful country and home. We thank Sylvain Pont from LMCM lab/MNHN Direction des collections MEB Service for MEM-EDS Imaging. Lastly we thank Mr. François Escuillié for the authorization to photograph the specimens in his collection.

Additional Information and Declarations

Competing Interests

Author Contributions

New Species Registration

The authors declare there are no competing interests.

André Nel and Romain Garrouste analyzed the data, wrote the paper, prepared figures and/or tables, reviewed drafts of the paper.

Patricia Nel and Régis Krieg-Jacquier analyzed the data, wrote the paper, reviewed drafts of the paper.

Jean-Marc Pouillon analyzed the data, reviewed drafts of the paper.

The following information was supplied regarding the registration of a newly described species:

Zoobank LSID: urn:lsid:zoobank.org:act:pub:8154DAD2-355D-4591-A4FC-7E6EFC95B368.

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
