# Peer review of "Exceptionally preserved insect fossils in the Late Jurassic lagoon of Orbagnoux (Rhône Valley, France)"

_PeerJ, doi:10.7717/peerj.510_

## Round 0.1 · original submission · Major Revisions

Congratulations with these interesting Upper Jurassic discoveries, including the oldest record of the heteropterand infraorder Gerromorpha. However, the manuscript still needs some major work before it can be published; several statements are made without citing adequate references. Furthermore, selling the Orbagnaux aite as a important Konservät-Lagerstätte is exaggerated considering only one insect fossil and some insect traces on leaves were found (see reviewer 2). Additionally, you mention it as an important calibration points for the origin of Mesoveliidae, but you provide very little dating evidence which are relevant to justify. For these reasons, I agree with reviewer 2 that major revisions are needed. However, I do feel these changes are easy and not too time-consuming to make. The main points which need to be addressed are:

1) Selling Orbagnaux as a Major Insect Konservät-Lagerstätte: You oversell this localities as a major future Konservät-Lagerstätte for insects, which is at the moment extremely exaggerated as only 1 insect and some traces on leaves have been found; Please focus on the exceptionally preserved gerromorpha specimen; you need to be more honest and speak of a potential major Lagerstätte in the discussion; change the title and text accordingly. I feel your observations and some interpretations could yield a sound paper (backed up by all reviewers) without the necessity to make overinterpretations of the data to make the paper more attractive.
2) Age of the Solnhofen and Orbagnaux site: you refer to the Solnhofen-Eichstätt plattenkalke as Kimmeridgian although some have now been dated as Tithonian (see further comments below; Reviewer 2)
3) Taxonomy and nomenclature of non-insect fossils: you should determine the plants, fish and shrimp fossils more precisely as these fossils are so well-preserved that more precise taxonomic determinations are possible, even by non-experts. Certaintly, when this site is so well-known as you claim for non-insect fossils.
4) Attributions of the traces to Chrysomelidae: The presence of Chrysomelidae is still debated; New molecular studies (Gómez-Zurita et al. 2007) indicate that the group might have not been present at that time, while other (Bo et al. 2014) recently made new finds and discussed their implications for the origin of Chrysomelidae (compare comments by reviewer 3); Both should be taken in consideration and discussed in their context.
5) Personal communication about additional Cretaceous Mesoveliidae: This paper has been published in the meantime and needs to cited instead of the personal communication (please cite; see comments below)
6) Attribution of the fossils to the Kimmeridgian: you need to specify precisely what evidence the Kimmeridgian age of this fossil intervals is based. Furthermore, if you sell this a potential calibration you should strive to date this fossil as precisely as possible and provided the kind of evidence it is based on (compare Parham et al. 2011) Reference: Parham, J. F., Donoghue, P. C. J., Bell, C. J., Calway, T. D., Head, J. J., Holroyd, P. A., ... & Benton, M. J. (2012). Best Practices for Justifying Fossil Calibrations. SYSTEMATIC BIOLOGY, 61(2), 346-359.
7) Language used in the manuscript: the finished manuscript needs to be read by a English native speaker before submission
8) Missing references: several important statements are made about stratigraphy, comparison with Solnhofen-Eichstätt, traces caused by extant insects; etc. without citing appropriate references; this needs to improved (examples are given below and report by reviewer 1)



Additional points which needs to be addressed:
Title: Please change the title to “Exceptionally preserved insect fossils in the Late Jurassic lagoon of Orbagnoux (Rhone Valley, France”

Abstract, p. 1, line 16: “lithoraphic” needs to be changed to “lithographic”

Abstract, p. 1, lines 23-24: “can come crucial Korservat-Lagerstättes” should be replaced with “can potentially become an important Konservat-Lagerstätte”

p. 2, line 7: “Late Kimmeridgian” please replace with “Late Jurassic” as the plattenkalk Lagerstätte of Bavaria are dated to the Upper Jurassic; around Solnhofen and Eichstätt these are dated to the Tithonian and not to the Kimmeridgian; please verify in Schweigert, G. 2007. Ammonite biostratigraphy as a tool for dating Upper Jurassic lithographic limestones from South Germany - first results and open questions. Neues Jahrbuch für Geologie und Paläontologie, Abhandlungen, 245 (1): 117-125.

p. 2, line 11: “significant scientific studies”: you need to cite more references (including studies on taphonomy, flora and fauna) to back this statement up; just citing a paper on microbial mats in the Cerin Formation is insufficient

p. 2, line 15: “Solenhofen-Eichstätt” needs to be replaced with “Solnhofen-Eichstätt”; furthermore please cite references who published on palaeoentofauna from Solnhofen-Eichstätt to demonstrate the rich palaeonentofauna

p. 2, line 17: you need to cite references who published on these plants remains

p. 2, line 25: arthropods are present needs to be replaced with “might be present”

p. 3, line 2: “can become in the next future one of the major insect outcrops” is exaggerated; please change it to “has the potential to become an important insect outcrop in the future”; next future is not correct, it should be near future
p. 3, line 25: “aptyci” is wrongly spelled and needs to be replaced by “aptychi”

p. 4, line 1, “An insect has been in these layers”: which layers do you mean; this entire discussion on stratigraphy is very unclear; you need to be more specific on which data (ammonoids, microfossils or other fossils, lithostratigraphy) gives constraints on the age of these fossils and how confident you are on this age

p. 4, line 11-13: What is the main evidence for the interpretations as a lagoon delimited by barrier reefs and islands ? Please cite references and/or briefly discuss

p. 4, line 14: you postulate that some fly larvae could live shallow brackish environments; but how can you rule out transport from a different environment: some (marginal) marine Lagerstätte contain well-preserved terrestrial arthropods (even larvae) which have been transported from the continents or islands; see Brauckmann, C, Gröning, E., 2008. A first record of Insecta from the LAte Jurassic sequence of the LAngenberg near Oker, Lower Saxony (Germany). Clausthal Geowiss 6, 45-48 for a similar scenario (see reviewer 1)


p. 4, line 22: please replace “is clearly a Hemiptera Heteroptera:” by “belongs to the Heteroptera (Hemiptera):”

p. 5, line 4: “now” could be replaced by “here” or is this done by Kraemer et al.,

p. 5, line 5: please replace “Solórzano Kraemer et al., pers. comm.” by “Solórzano Kraemer et al. 2014) as the paper describing these fossils has been published in the meantime: Solórzano Kraemer, M.M., Perrichot, V., Soriano, C., Damgaard, J., 2014. Fossil water striders in Cretaceous French amber (Heteroptera: Gerromorpha: Mesoveliidae and Veliidae). Syst Entomol 39, 590-605.

p. 5, line 14: “included genera”: fossil or recent ?

p. 6, line 6: 60 mm or 6.0 mm ?

p. 6, line 6: shrimp is a not precisely defined term; you need to add a latin name of the taxon (be as precise as possible; otherwise use open nomenclature)

p. 7, line 4-6: “Unfortunately, the cephalipac trichobothria … above the eye” Why (too delicate, etc.) could these not be preserved ? Does their absence alter the interpretation of this taxon as Gerromorpha

p. 7, line 12: Replace “Absent Gallomesovelia” by “not observed or preserved in Gallomesovelia”

p. 7, line 25: Please rephrase “fossilisation in lateral view”: consult with a English native speaker as it is not entirely clear what you mean

p. 8, lines 6-10: These sentences can be merged as they gave almost identical information to “A further important difference is the great size of Gallomesovelia (body axis 6.0 mm long) when compared to modern and fossil Mesoveliidae (Mesoveloidea peruviana Drake, 1949: 2.9 mm; Madeovelia guineensis Poisson 1959: 2.6 mm)”; There should be more data available in the literature on body size of Mesovellidae. Please cite more.

p. 8, line 11-12: This should be corrected to “The Mesoveliidae are considred to be the most basal lineage of Gerromorpha (Damgaard, 2008; Weirauch & Schuh, 2012)”. Furthermore, Weirauch & Schuh 2012 is absent from the reference list. Please make sure that all references are listed

p. 8, line 18-19: you sell your fossil as a new landmark for dating gerromorphan lineages; However, you do provide all necessary information to make this a good fossil calibration (particularly the stratigraphic assignment and age of these fossils are poorly described).

p. 9, line 13 and line 16: “if” needs to be replaced with “Although”

p. 9, line 14: “100 leaves of Zamites”: from where do these derive; other localities in the same interval; please specify

p. 10, line 7: please replace attack with “damage”

p. 10, line 14: you need to cite reference which discuss that similar attacks on modern angiosperm leaves can be caused by larvae of Diptera, Lepidoptera or Coleoptera. This is one of the many examples where statements are made which are not back up by appropriate references.

p. 10, line 15: Chrysomelids are mentioned to be probably responsible for the damage. However, some molecular studies (Gómez-Zurita et al. 2007) reveal that this group could not be present and that all the older records are not reliable.
Gómez-Zurita, J., Hunt, T., Kopliku, F., Vogler, A.P., 2007. Recalibrated tree of leaf beetles (Chrysomelidae) indicates independent diversification of angiosperms and their insect herbivores. PLoS ONE 2, e360.
Recently, other have provided insights in the early evolution of Chrysomelids:
Wang Bo, Ma Junye, McKenna D., Yan E.V., Zhang Haichun, Jarzembowski E.A. (2014) The earliest known longhorn beetle (Cerambycidae: Prioninae) and implications for the early evolution of Chrysomeloidea. Journal of Systematic Palaeontology, 12: 565–574. Both studies and their implications should be at least taken in consideration and discussed.

p. 10, line 21: please be more specific; use latin names; which “fishes”, “shrimps”, “worms”

p. 11, line 2: please add “as postulated by Tribovillard et al. (1999)” behind “was clearly not abiotic”

p. 11, line 5-6: it would be helpful to figure some marine fossils to back up these statements or at least cite some previous studies who have reported them; furthermore, some groups like ammonites could potentially be used to date these layers more precisely (compare Schweigert et al. 2007).

p. 11, line 17: Bavarian “Kimmeridgian”; you need to be more specific and this is wrong when you are speaking about the localities around Solnhofen and Eichstätt as pointed out by reviewer 1: compare Schweigert, G. 2007. Ammonite biostratigraphy as a tool for dating Upper Jurassic lithographic limestones from South Germany - first results and open questions. Neues Jahrbuch für Geologie und Paläontologie, Abhandlungen, 245 (1): 117-125.
http://dx.doi.org/10.1127/0077-7749/2007/0245-0117

p. 11, line 24: this is not a popular book; please change “new insects world” to “new insect habitat”

p. 17, figure 1: the “fern”, “fish” and “fish” are well-enough preserved to be determined more precisely, certainly if this is a well-investigated locality for plants and fishes as you claim in the introduction; it should be easy to ask some specialists to make more precise determinations as some of you are working in the natural history museum of Paris which houses experts on these groups (e.g., Philippe Janvier for fishes; etc.). Use at a minimum open nomenclature with latin names for the “shrimp”

p. 17, figure 3: as also discussed by reviewer 2, the MEB image does not provide any more information than the photographs, so it should be remade or potentially left away.

Figures in general: scale-bars should be consistently placed in the lower right corner; abbreviations need to be explained in the captions (see comments reviewer 1)

The additional comments/suggestions by the reviewers, particularly those by reviewer 1, also need to be addressed and integrated.

Reviewer 1 ·

Basic reporting

See attached PDF.

should be read by a native speaker before submission

Experimental design

See attached PDF.

Validity of the findings

See attached PDF

These findings are really spectacular and interesting but based on only one single specimen and herbivory traces on a few leaves the general interpretation for a new extraordinary Konservat Lagerstätte should be expressed in a bit more conservative way

Annotated reviews are not available for download in order to protect the identity of reviewers who chose to remain anonymous.

·

Basic reporting

No Comments

Experimental design

No Comments

Validity of the findings

No Comments

Additional comments

This good contribution to paleontology which should be published. I have only small remarks on this (see the attached ms)

Reviewer 3 ·

Basic reporting

No Comments

Experimental design

No comments

Validity of the findings

No Comments

Additional comments

This is potentially a significant new finding. The paper is well written and the description is correct. The images are good, and show beautiful specimens. I recommend acceptance with minor revisions.
I made some comments directly on the attached document and would like to give here only one comment. Please provide a geographic map of the fossil localities. It will be very helpful for readers.

Annotated reviews are not available for download in order to protect the identity of reviewers who chose to remain anonymous.

---

## Round 0.2 · accepted · Accept

Thank you for integrating all our suggestions. I am pleased to see a publication on this interesting fossil locality. I still welcome you to to integrate your hypothesis that some of these traces were produced by Chrysomelidae. You could for example still add the following sentence "These structures are similar to traces produced by Chrysomelidae, but the presence of Chrysomelidae during the Jurassic is still debated (compare Gómez-Zurita et al. 2007; Bo et al. 2014).", which would make your manuscript even more interesting.

Reviewer 1 ·

Basic reporting

see notes below

Experimental design

see notes below

Validity of the findings

see notes below

Additional comments

see notes below